# An Overlooked Challenge: A Retrospective Audit of Overnutrition in Hospital Rehabilitation Wards

**DOI:** 10.3390/healthcare13020188

**Published:** 2025-01-18

**Authors:** Hannah T. Olufson, Jennifer Ellick, Simone McCoy, Sally E. Barrimore, Tracy Knowlman, Adrienne M. Young

**Affiliations:** 1Dietetics & Food Services, Surgical, Treatment & Rehabilitation Service (STARS), Metro North Health, Brisbane, QLD 4029, Australia; 2STARS Education & Research Alliance, STARS, University of Queensland & Metro North Health, Brisbane, QLD 4029, Australia; 3Nutrition Research Collaborative, Royal Brisbane & Women’s Hospital, Metro North Health, Brisbane, QLD 4029, Australia; 4Dietetics & Food Services, Prince Charles Hospital, Metro North Health, Brisbane, QLD 4032, Australia; 5Dietetics & Food Services, Redcliffe Hospital, Metro North Health, Brisbane, QLD 4020, Australia; 6Centre for Health Services Research, University of Queensland, Brisbane, QLD 4067, Australia; 7Dietetics & Food Services, Royal Brisbane & Women’s Hospital, Metro North Health, Brisbane, QLD 4029, Australia

**Keywords:** health care, dietetics, food services, rehabilitation, overnutrition, malnutrition, clinical audit

## Abstract

**Background/Objective:** Research shows that obesity has risen among rehabilitation patients. Despite this, nutrition care in subacute rehabilitation wards focuses primarily on preventing and treating protein-energy malnutrition. The continued provision of energy-dense meals during lengthy rehabilitation admissions may present a risk of overnutrition for some patients, which can adversely affect functional outcomes. However, overnutrition is not routinely monitored in practice. This study summarizes the initial findings of a multi-site investigation of overnutrition incidence across five rehabilitation wards to scope the need for future research. **Methods:** A retrospective audit was conducted, including all inpatients admitted over 3 months to the study wards with a complete dataset (total sample *n* = 199). Data were collected from the medical record and menu management system to determine overnutrition, defined as an average daily energy intake equal to or greater than 1000 kJ above estimated requirements and weight gain of equal to or greater than 1 kg over the admission. **Results:** The incidence of overnutrition in the total sample was 12.1%. Of those patients deemed at low risk of malnutrition (*n* = 124), 19.4% developed overnutrition during their rehabilitation admission. Those who developed overnutrition during their admission gained an average of 2.9 kg, with a mean excess energy intake of 2456 kJ/day above estimated requirements. They also consumed a high intake of discretionary items (mean of 3156 kJ/day). **Conclusions:** The findings suggest that further research is needed to investigate the etiology and impact of the overlooked problem of overnutrition in subacute rehabilitation settings. Future investigation is essential to ensure that the planning and delivery of subacute dietetic and food services meet the nutrition needs of patients in longer-stay inpatient settings.

## 1. Introduction

International guidelines for nutrition care in hospitals support the design of nutrition and food services that prevent and treat protein-energy undernutrition (herein referred to as ‘malnutrition’). Malnutrition is an important problem in acute hospitals and persists in rehabilitation, with prevalence rates of 30–50% [1]. With increased recognition of malnutrition and its impact on rehabilitation outcomes [1], food services have been primarily designed to promote increased nutritional intake, including providing high-energy, high-protein meals and snacks, fortified foods, and supportive mealtime environments [2,3,4,5]. However, the high availability of these energy-dense foods and beverages that often have low nutritional value (e.g., chocolates and cakes) may risk overnutrition for some patients, negating secondary prevention efforts and a return to healthy eating.

Demand for rehabilitation has risen worldwide and is predicted to increase further as the global population continues to age and the prevalence of chronic and complex disease grows [6,7]. Subacute inpatient rehabilitation may be offered to patients after an acute hospital admission to improve functioning prior to discharge into the community [8], including patients with neurological, musculoskeletal, respiratory, and cardiac conditions [9]. Dietitians working in subacute rehabilitation wards deliver interventions to target not only malnutrition and ongoing nutritional issues following acute illness but also and, typically less frequently, the secondary prevention of diet-modifiable conditions such as stroke and management of chronic disease [10,11].

In line with global trends, research suggests that the prevalence of obesity has risen considerably among rehabilitation patients [12], with a recent study showing that 28% of patients were living with obesity on admission to rehabilitation [13], with similar findings by in other inpatient settings [14]. Obesity has been associated with longer and more costly hospital admissions, higher post-surgery complications, impaired wound healing, increased rates of infection, and a greater risk of morbidity and mortality [14]. Additionally, people with sarcopenic obesity, where sarcopenia and obesity co-exist, may have a higher risk of functional impairment and cardiometabolic comorbidities than either condition alone [15]. Consequently, the prevention and treatment of overweight and obesity remain a priority across healthcare settings globally.

The existing literature focused on rehabilitation patients with spinal cord injuries emphasizes that excessive energy intake can lead to overnutrition, which adversely affects functional outcomes [16,17,18]. This may be exacerbated by the high availability of fortified foods and discretionary items (i.e., energy-dense foods and beverages with low nutritional value) in hospital food service systems designed to prevent and treat malnutrition. Therefore, to ensure the appropriate provision of nutrition and food services in the inpatient settings, including subacute rehabilitation wards, it is necessary to undertake quality assurance activities that investigate rates of both malnutrition (undernutrition) and overnutrition regularly. Annual point-prevalence audits of malnutrition (undernutrition) are regularly conducted in hospitals across different countries with the aim of measuring, comparing, and ultimately reducing malnutrition prevalence; an example of this is the annual nutrition Day audits conducted across Europe, North America, and parts of Asia [19]. However, to our knowledge, similar audits focused on overnutrition (i.e., excess energy consumption) are not regularly undertaken in healthcare settings. These audits are important in longer-stay settings, especially, such as subacute rehabilitation wards, where malnutrition and overnutrition may be concurrent nutritional concerns.

This study was a retrospective audit of overnutrition in five subacute rehabilitation wards to investigate this potential problem raised by local staff and consumers and determine the need for further dedicated research. A secondary aim was to explore dietary behaviors in patients with overnutrition.

## 2. Materials and Methods

A retrospective multi-site clinical audit of overnutrition (referred to herein as an ‘overnutrition audit’) was completed across five subacute general rehabilitation wards (i.e., excluding specialist spinal or brain injury rehabilitation or acute stroke beds) in three hospitals in Queensland, Australia. Patients are generally admitted to these rehabilitation wards following an acute medical or surgical admission in hospital wards and/or nearby referral hospitals.

Wards A, B, and C were all within one hospital with a combined 72 general rehabilitation beds (30, 30, and 12 beds, respectively). Ward D and E were both rehabilitation wards embedded within two separate acute care hospitals, with 24 and 29 general rehabilitation beds, respectively. Further details regarding the food service system and local malnutrition- and overnutrition-focused strategies in each ward are available in Table 1.

A convivence sampling approach was used, whereby all patients admitted and discharged within a pre-defined 3-month period between August 2022 to February 2023 (inclusive) were eligible for inclusion in the retrospective audit. To explore overnutrition incidence, only those patients at low risk of malnutrition were included; patients were excluded if underweight [20] or at risk of malnutrition (Malnutrition Screening Tool score ≥2, with reported unintentional weight loss [21]) according to routine admission screening by nurses. Patients returning a score of ≥2 were included in the audit only if their MST response was “unsure” of weight loss and “no” to the reduced intake question. Patients were also excluded if eligibility or undernutrition could not be established (i.e., missing MST or weight data).

In the absence of a published definition of overnutrition in rehabilitation, the project team (comprising experienced accredited practicing dietitians) developed multi-component criteria aligned with the well-accepted definition of malnutrition, referring to a deficiency or excess of energy, protein and other nutrients leading to adverse effects on body form (body shape, size, composition), function and outcome [4]. We defined overnutrition as “excessive energy intake equal to or greater than 1000 kJ/d above estimated energy requirements plus unintentional weight gain of equal to or greater than 1 kg during the admission”. Weight gain equal to or greater than 1 kg was decided to be clinically meaningful and unlikely to reflect measurement inaccuracy when considered in the context of energy intake equal to or greater than 1000 kJ/d more than estimated energy requirements. Estimated energy requirements were determined using the ratio method of 100 kJ/kg of body weight, as per local practice informed by evidence-based guidelines [22], with a maximum target of 8000 kJ/day. The protocol and tools for the overnutrition audit were piloted in Wards A and B in 2021, with refinements made to the scope and focus of data collection.

**Table 1 healthcare-13-00188-t001:** Overview of nutrition and food service systems on the included wards.

Ward	Beds (*n*)	Food Service System	Malnutrition-Focused Strategies	Overnutrition-Focused Strategies	Intake Tracking Functionality
**Ward A**	30	Breakfast: central pre-plating Lunch and dinner: bulk decentralized trolley service with meals provided in a communal dining room Snacks: - Ward A and B: on-demand snacks - Ward C: tea-trolley service Default menu: standard diet	- Routine malnutrition screening - Automatic delegation of “at-risk” patients to a dietetic assistant for assessment and management, with dietitian supervision (and additional input as needed) [23] - Patient education groups on nutrition support strategies - On-demand snacks - Dedicated staff roles in the dining room to support mealtime care	- Patient education groups on healthy eating - Staff education for dietitians about overnutrition risk - Dietitian assessment/intervention on referral	Yes—intake tracking of all meals and snacks provided by hospital food service in the electronic meal management system (CBORD, Roper Technologies)
**Ward B**	30				
**Ward C**	12				
**Ward D**	24	Breakfast, lunch, and dinner: central pre-plating food service model Snacks: tea-trolley service Default menu: high-protein, high-energy	- Routine malnutrition screening - Dietitian referral for patients screened as “at-risk” of malnutrition	- Dietitian assessment/intervention on referral	Yes—functionality available in each site’s menu management system but not yet implemented
**Ward E**	29	Breakfast, lunch, dinner, and snacks: room service food service model with on-demand ordering via a smart-phone application or call center Default menu: high-protein, high-energy room service menu			

Demographic (age, gender) and anthropometric data (admission and discharge weight, height) were collected from the medical record. Energy and discretionary food and beverage intake were calculated from intake records for all meals and between-meal snacks in Wards A, B, and C. In the absence of intake tracking data for Wards D and E, meal orders were used as a proxy for intake, with the assumption that 100% of orders were consumed. Patients were excluded if there were fewer than five days of intake or ordering information available. Known food composition data provided by suppliers was used to determine the nutritional composition of menu items. The diet code (i.e., type of menu offered to the patient) and intake of discretionary menu items (as per national guidelines [24]) were also recorded from intake/ordering records. All data were collected by final-year postgraduate dietetics students under the supervision of a local dietitian. The primary author oversaw data cleaning and analysis and revisited primary data sources to cross-check the data extraction and ensure data quality.

Data were entered and managed in REDCap (Research Electronic Data Capture) [25,26]. The data were analyzed in Microsoft^®^ Excel (Version 2412) and REDCap using descriptive statistics (with histograms created to assess normality) to summarize population demographics, overnutrition rates, and explore energy and discretionary food intake.

The hospital human research ethics committee chairperson approved the protocol as a quality assurance audit (EX/2022/MNHB/90168). As approved by the ethics committee chairperson, the need to obtain patient consent was waived as the study used only routinely collected data from hospital records. This study conforms to all STROBE guidelines and reports the required information accordingly [27].

## 3. Results

Of the 248 patients admitted to study wards during the study period, 199 patients had complete datasets and were therefore eligible for inclusion in the audit. Of these 199 patients, 75 deemed at risk of malnutrition were excluded during screening, resulting in 124 patients being included in the overnutrition analyses. Further details regarding exclusion reasons are available in Figure 1. For the 124 patients included in the audit, the median age was 75 years (IQR 64.8–81.0 years), and the median length of stay was 20 days (IQR 13.8–28.0 days). The median BMI for the 124 included patients was 28.5 kg/m^2^ (IQR 25.2–33.7 kg/m^2^), 30 patients had an admission BMI between 18.5–24.9 kg/m^2^, 44 between 25.0–29.9 kg/m^2^, 25 between 30.0–34.9 kg/m^2^, 13 between 35.0–39.9 kg/m^2^, and 12 had an admission BMI ≥ 40.0 kg/m^2^. Further details regarding the characteristics of the included patients are available in Table 2.

Overall, 24 patients developed overnutrition during their rehabilitation. The characteristics of the participants who experienced overnutrition are summarized in Table 3. This was an overnutrition incidence of 12.1% of the total sample (*n* = 199) and 19.4% of those at low risk of malnutrition (*n* = 124). Patients who experienced overnutrition consumed a mean of 9414 kJ/d (standard deviation (SD) 1107 kJ/d), equating to an excess intake of 2456 kJ/d (SD 1344 kJ/d), and they gained a mean of 2.9 kg (SD 2.9 kg). Nine patients gained weight (but did not have ‘excessive energy intake’ from the food and beverages provided by the hospital food service); these patients gained a mean of 4.5 kg (SD 2.9 kg) during their rehabilitation admission.

Of those who experienced overnutrition (*n* = 24), 12 (50%) were on a ‘standard’ diet at discharge, and 7 (29%) were on a ‘high-protein, high-energy’ diet on admission to the rehabilitation ward, with the remainder on an ’easy-chew’ diet (*n* = 2), a combined ‘high-protein, high-energy easy-chew’ diet (*n* = 2), and a ‘diabetic’ and ‘high-protein, high-energy’ diet (*n* = 1). Patients with overnutrition consumed a median of 5.3 serves/day of discretionary items (IQR 3.7–5.9 serves/day), contributing a mean of 3156 kJ/d (SD 1305 kJ/d) to their total energy intake.

## 4. Discussion

This audit revealed an overnutrition incidence of 12.1% of the total sample (19.4% of patients at low risk of malnutrition). These patients gained a clinically meaningful amount of weight during their rehabilitation admission (an average of 2.9 kg), with evidence of excess energy intake above their estimated requirements (mean excess energy intake of 2456 kJ/day). These initial findings indicate that dedicated research is warranted to explore the etiology and impact of this traditionally overlooked nutritional problem in rehabilitation. Additionally, for dietitians working in longer-stay inpatient settings, such as subacute rehabilitation wards, these results may raise the importance of considering overnutrition in addition to malnutrition in nutrition and food service delivery and design.

There are limited studies exploring overnutrition in patients admitted to subacute settings, such as rehabilitation wards. Most research and nutritional recommendations citing overfeeding in hospitalized patients are largely focused on the critically ill [28,29], who have significantly different nutritional and medical needs to those of rehabilitation patients. However, others have emphasized the risk of overprescribing energy intake in people with spinal cord injury, exacerbating unintentional weight gain, which, in turn, may reduce mobility and independence and increase rates of rehospitalization [16]. For patients undergoing rehabilitation, carrying excess weight may impact functional ability [29], with research suggesting that patients living with obesity tend to face greater medical complexity and reduced function in comparison to those without obesity [12,30]. Overweight and obesity also incur significant healthcare and hospital costs [31,32], including increased staff labor [33]. In the rehabilitation setting, this may be reflected in increased staff labor associated with reduced patient mobility, such as for patient transfers or physical therapy. Thus, the findings from this overnutrition audit require further investigation, as others have also recently suggested [14], focusing on including body composition measurements to differentiate fat and muscle mass changes, which was outside the scope of this retrospective study.

The co-existence of malnutrition and overnutrition is a challenge experienced in other non-acute settings such as mental health, with previous research suggesting that the easy access of high-energy foods and drinks contributes to the obesogenic nature of inpatient mental health units [34]. In their study, Faulkner et al. also suggested a review of food service arrangements for hospitals, positing that one hospital food service system does not necessarily fit the needs of all patients, i.e., across different wards [34]. A similar sentiment was echoed by Flint and colleagues [35], who suggested that systems which support the provision of high-protein, high-energy main meals and snacks need to be carefully implemented in inpatient mental health settings, where overfeeding and unintentional weight gain are a concurrent concern to malnutrition. Our findings may suggest that this same consideration extends to the rehabilitation setting, where discretionary items, often in abundance on high-protein, high-energy menus to treat malnutrition, contributed on average 3156 kJ/day to the nutritional intake of patients who developed overnutrition.

The patients included in this audit who developed overnutrition during their rehabilitation admission consumed a high number of discretionary items, with a median intake of 5.3 serves/day. Dietary guidelines worldwide recommend limiting discretionary food and drinks that are high in energy, fat, salt, and sugar and low in essential nutrients [36,37,38,39], with the Australian Dietary Guidelines suggesting a limit of 0–3 serves of discretionary serves daily for men and 0–2.5 serves daily for women [40]. This is recommended due to the higher energy, added sugar, added salt, and saturated fat in discretionary foods and drinks, which have been associated with an increased risk of conditions such as obesity, type 2 diabetes, heart disease, stroke, and some cancers [39,40]. Consequently, our findings may indicate a need and opportunity for dietitians in longer-stay healthcare settings to address excessive discretionary food and drink intakes, which can displace the consumption of more nutritious foods that better support rehabilitation and recovery.

Furthermore, research exploring food service arrangements and systems that meet the diverse nutrition needs of patients in rehabilitation is lacking. Our initial findings suggesting an overnutrition incidence of 19.4% in rehabilitation patients deemed at low risk of malnutrition may warrant further research into if and how food service systems, menu standards, and models of nutrition care in subacute settings should be designed to support patients with opposing nutritional needs (overnutrition and malnutrition). This may include developing new approaches to nutrition screening to identify patients who may be at risk of overnutrition, as well as co-designing options for nutrition intervention. Co-design is yet to be undertaken in the rehabilitation setting and would be valuable in this context to engage end-users in designing services aimed at better meeting the diverse nutritional needs of patients in this setting. Additionally, research using a prospective cohort study design may be beneficial to investigate overnutrition incidence (including considering body composition via bioelectrical impedance analysis and/or examining other clinical biomarkers) and influential factors in greater detail to inform future improvements in subacute care.

There are limitations with this study related to its reliance on data collected and documented in routine practice, as well as intake or ordering data that only capture food and beverages provided by the hospital. Of note is the fact that nine patients gained weight (≥1 kg) during their admission but had not consumed excessive amounts of the foods and fluids provided by the hospital food service. It is possible that these patients were consuming excessive amounts of foods and beverages from other sources (brought in by family members, ordered through meal delivery services, or sourced from the hospital cafeteria or vending machines), which is anecdotally known to occur and may explain this weight gain. Furthermore, changes in body composition were not measured in this audit, a focus for future research. A strength of this study was the consideration of multiple parameters to determine overnutrition, rather than relying on BMI alone. In the absence of intake data at two sites, ordering data were used as a proxy measure, assuming that all of what was ordered by patients was consumed. This was reinforced by the fact that all patients included were screened as having a good appetite (as per their MST score) and the inclusion of weight gain in the overnutrition criteria. Given the small sample size, it is difficult to draw conclusions related to how specific features of the dietetics and food service systems across the three sites influenced overnutrition; however, this should be explored in future studies.

## 5. Conclusions

The initial findings from this study revealed that 12.1% of all rehabilitation patients included in the audit sample developed overnutrition. Those who developed overnutrition during admission gained an average of 2.9 kg, with a mean excess energy intake of 2456 kJ/day above estimated requirements. They also consumed a high intake of discretionary items (mean of 3156 kJ/day). These results need to be confirmed in larger prospective studies, including measures of body composition. However, this study is the first step in establishing the need for further research into the etiology and impact of overnutrition in rehabilitation wards. Future research should be complemented by collaborative efforts to inform the design of suitable nutrition and food service systems for longer-stay inpatient settings where both malnutrition and overnutrition may co-exist.

## Figures and Tables

**Figure 1 healthcare-13-00188-f001:**
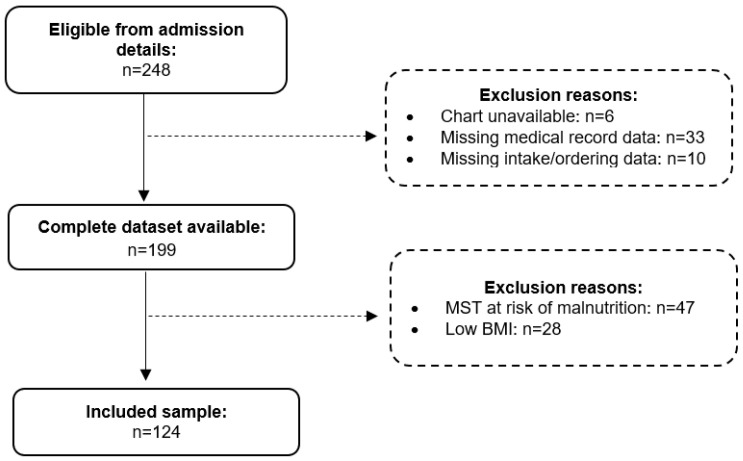
The flowchart for the overnutrition screening and audit processes.

**Table 2 healthcare-13-00188-t002:** Demographic details of all patients included in the audit sample.

Variable	Included Sample (*n* = 124)	Patients Who Developed Overnutrition (*n* = 24)	Patients Who Did Not Develop Overnutrition (*n* = 100)
Age (years) ^a^	75.0 (64.8–81.0)	76.5 (66.3–84.0)	75.0 (64.8–80.0)
Length of stay (days) ^a^	20.0 (13.8–28)	23.0 (14.0–30.3)	19.5 (13.0–28.0)
Gender (% male)	50.0	58.3	48.0
Admission BMI (kg/m2) ^a^	28.5 (25.2–33.7)	24.8 (22.7–28.0)	29.3 (26.2–34.6)

^a^ Data represented as median (interquartile range).

**Table 3 healthcare-13-00188-t003:** Additional characteristics and dietary information of patients with overnutrition (*n* = 24).

Variable		Results for Patients with Overnutrition
Weight gain (kg) ^a^		2.9 (2.9)
Energy intake (kJ/day) ^a^		9414 (1107)
Energy intake above estimated requirements (kJ/d) ^a^		2456 (1344)
Discretionary serves consumed (serves/day) ^b^		5.3 (3.7–5.9)
Admission diet codes ^c^	Standard diet High-protein, high-energy diet Easy-chew diet Easy-chew + high-protein, high-energy diet Diabetic + high-protein, high-energy diet	12
		7
		2
		2
		1

^a^ Data represented as mean (standard deviation) for weight gain and energy intake above estimated requirements. ^b^ Data represented as median (interquartile range). ^c^ Data represented as the total number of patients (*n*) on each diet code on admission to rehabilitation.

## Data Availability

The data presented in this study are available on request from the corresponding author.

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
