# Peer review of "An Overlooked Challenge: A Retrospective Audit of Overnutrition in Hospital Rehabilitation Wards"

_healthcare, 2025, doi:10.3390/healthcare13020188_

Round 1
Reviewer 1 Report
Comments and Suggestions for Authors
The paper have lack of information in results and discussion.
Methodology could be improved to a abetter understanding of the research.
This points are essential to the publication of the paper.
Reviewer 2 Report
Comments and Suggestions for Authors
Please see Attached file for comments.

Reviewer 3 Report
Comments and Suggestions for Authors
The content of this manuscript is very interesting and important.
I would suggest some improvements:
Abstract: Introduce with a few words in the background why this issue is relevant. The first sentence of the results is not very clear. Explain the results better. Those with overnutrition (overnutrition at t0 or those who gained the weight in hospital?).
Introduction: add some information on overweight and obesity as risk factors during hospitalization and for free living. Add something about sarcopenic obesity.
Results: add more results. Add a table with information at t0. It is important to know the % of patients that were already overweight/obese at t0. Present more clearly the results.
Line 237: the information about external sources is important and interesting. Can you add some information on that.
Best wishes
Round 2
Reviewer 1 Report
Comments and Suggestions for Authors The corrections made by the authors are in accordance with those requested.
Reviewer 2 Report
Comments and Suggestions for Authors
No More Comments